# LUSTER: Link Prediction Utilizing Shared-Latent Space Representation in Multi-Layer Networks

## Abstract

Link prediction in multi-layer networks is a longstanding issue that predicts missing links based on the observed structures across all layers. Existing link prediction methods in multi-layer network typically merge the multi-layer network into a single-layer network and/or perform explicit calculations using intra-layer and inter-layer similarity metrics. However, these approaches often overlook the role of coupling in multi-layer networks, specifically the shared information and latent relationships between layers, which in turn limits prediction performance. This calls the need for methods that can extract representations in a shared-latent space to enhance inter-layer information sharing and prediction performance. In this paper, we propose a novel end-to-end framework namely: **L**ink prediction **U**tilizing **S**hared-la**T**ent spac**E** **R**epresentation (LUSTER) in multi-layer networks. LUSTER consists of four key modules: the representation extractor, the latent space learner, the complementary enhancer, and the link predictor. The representation extractor focuses on learning the intra-layer representations of each layer, capturing the data characteristics within the layer. The latent space learner extracts representations from the shared-latent space across different network layers through adversarial training. The complementary enhancer combines the intra-layer representations and the shared-latent space representations through orthogonal fusion, providing comprehensive information. Finally, the link predictor uses the enhanced representations to predict missing links. Extensive experimental analyses demonstrate that LUSTER outperforms state-of-the-art methods for link prediction in multi-layer networks, improving the AUC metric by up to 15.87%.

## CCS Concepts

• **Computing methodologies** → **Neural networks**.

## Keywords

link prediction, multi-layer networks, shared-latent space, adversarial training, orthogonal fusion

**ACM Reference Format:**
Anonymous Author(s). 2024. LUSTER: Link Prediction Utilizing Shared-Latent Space Representation in Multi-Layer Networks. In . ACM, New York, NY, USA, 12 pages. https://doi.org/10.1145/nnnnnnn.nnnnnnn

## 1 Introduction

In recent years, link prediction for complex networks has attracted significant research attention [13, 31, 51]. Complex networks refer to systems with intricate structures, high heterogeneity or rich hierarchical levels [9], such as social relationships [20], transportation [26], and Internet structures [52]. These networks may not necessarily encompass a large number of nodes and edges, however, the relationships between nodes often exhibit diversity and complexity, making the modeling and analysis of these networks critically important [42]. Different types of complex networks include: heterogeneous networks [34], temporal networks [47], and multi-layer networks [19], each suited for handling complex information in different scenarios. Amongst them, multi-layer networks are regarded as an effective approach for processing multi-dimensional and multi-level information within complex networks [44]. In multi-layer networks, the overall structure is divided into multiple layers, each capturing a specific type of relationship, thus providing a comprehensive representation of the interactions and information exchange between node entities [8, 12, 23]. For example, in a transportation system, aviation, railways, and highways can be regarded as distinct layers, each of which describes the connection between cities through different modes of transportation.

Existing research for link prediction in multi-layer networks usually focus more on merging the multi-layer network into a single-layer network [33, 43], and/or extracting structural features of each layer using multiple similarity metrics [41]. However, these methods fail to fully incorporate the role of the coupling in multi-layer networks, *i.e.,* the shared information and latent relationships between layers [10], which in turn limits the potential of multi-layer networks for the prediction tasks. Therefore, it is necessary to develop methods that can better extract representations from a shared-latent space to enhance inter-layer coupling and improve prediction performance.

Previous studies [7, 25, 53] have demonstrated there exists a shared-latent space between different data sources, yet its potential has not been fully exploited for multi-layer network analysis. To fully consider the coupling in multi-layer networks, we treat links from different layers as data originating from different sources in this work to extract representations from the shared-latent space. By combining these representations with intra-layer representations, we capture the complex structures features within each layer, while identifying and leveraging inter-layer coupling, thus improving prediction performance. An example illustration in this regard is shown in Fig. 1, which shows a three-layer transportation network. It highlights that although the interactions between nodes in the aviation, railway, and highway layers are different, yet these layers are not completely isolated. For this, by utilizing the inter-layer coupling, i.e., the mutual influence and connections between different layers, there may be a high probability link between San

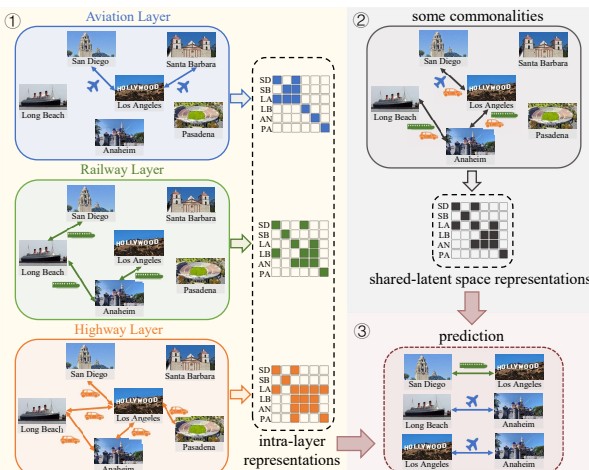

**Figure 1: An example of a three-layer transportation network: combining intra-layer representations and shared-latent space representations for prediction.**

Diego and Los Angeles in the railway layer, based on their existing connections in the aviation layer and similar patterns in the highway layer.

However, to effectively combine the intra-layer representations and shared-latent space representations for effective modeling of multi-layer network, we foresee following key challenges. The *first* challenge is to identify the shared-latent space. It is relatively challenging to track due to the dynamic nature and high-dimensionality of the shared-latent space. The *second* challenge is how to combine the intra-layer representations and the shared-latent space representations for link prediction tasks. The shared-latent space representations are novel representations derived from multiple intra-layer representations, which integrates features from different layers into a higher-level feature expression. There is a need for effective combination to avoid linear dependencies.

To address these challenges, in this work we propose a novel framework namely: **L**ink prediction **U**tilizing **S**hared-la**T**ent spac**E** **R**epresentation (LUSTER) for effective modeling of multi-layer networks. LUSTER primarily encompasses four key components: (i) Representation Extractor, (ii) Latent Space Learner, (iii) Complementary Enhancer, and (iv) Link Predictor. *"Representation Extractor"* is responsible for learning the intra-layer representations using the structural information of each layer. *"Latent Space Learner"* aims to extract representations from the shared-latent space through adversarial training in order to effectively dig-out the inter-layer coupling. Later, *"Complementary Enhancer"* uses orthogonal fusion to organically combine the intra-layer representations and the shared-latent space representations to further improve the quality of feature representations. Finally, *"Link Predictor"* performs the link prediction task based on the enhanced representations to accurately determine potentially missing links. We argue the proposed model not only captures the internal details of each layer, but also improves the ability to capture the latent shared relationships between different layers, thereby improving the prediction performance of the end-model.

We summarize the key contributions of this work as follows:

- We propose LUSTER, a novel method that integrates intra-layer representations and shared-latent space representations to better account for the inter-layer coupling.
- We design adversarial training to obtain shared-latent space representations across different layers and use orthogonal fusion to combine these representations with intra-layer representations, ensuring minimal redundancy.
- We conduct extensive experiments to demonstrate that LUSTER outperforms state-of-the-art models for link prediction in multi-layer networks by improving the AUC metric by up to 15.87%.[1]

## 2 Related Work

We bifurcate the existing work into: (i) Link prediction in multi-layer networks, (ii) Shared-latent space and (iii) Adversarial neural networks.

### 2.1 Link Prediction in Multi-Layer Networks

Many existing link prediction models have been applied to multi-layer networks. Najari et al. [35] comprehensively considered the intra-layer similarity and representations extracted from the prediction layer. Abdolhosseini et al. [1] utilized the structural representations of other layers for the optimal reconstruction of target layer structure. In addition, Luo et al. [27] proposed a new multi-attribute decision making method which defines a layer similarity measure based on cosine similarity to achieve the weighting of each layer. Mandal et al. [30] reported that the quality of feature group selection significantly influences the effect of deep-learning models. However, the traditional topology calculation methods mentioned above exhibit limitations in terms of flexibility and efficiency.

In recent years, deep learning models have excelled in various link prediction tasks owing to their inherent feature extraction capabilities. Yao et al. [54] proposed a node similarity index based on layer relevance by utilizing the intra-layer and inter-layer representations. Shan et al. [41] extracted a set of elaborate structural representations of links from all layers. In addition, Mishra et al. [32] combined information from multiple layers into a single weighted network, accounting for the relative density of each layer. They proposed MNERLP algorithm, which first calculates node and edge relevance based on the summarized graph, and then combines both these factors to perform link prediction. After that, Mishra et al. [33] proposed HOPLP algorithm which iteratively calculates link likelihoods taking longer paths between nodes into account. However, these methods often focus on learning intra-layer local representations and fail to fully exploit the shared information across layers. This can result in conflicting predictions for links in different layers. Therefore, we aim to extract representations from a shared-latent space to capture cross-layer information, thereby enhancing the accuracy and consistency of predictions.

### 2.2 Shared-Latent Space

Shared-latent space is a unified feature space that integrates information from diverse data sources to capture potential relationships

---

[1]The code is available at an anonymous repository: https://anonymous.4open.science/r/LUSTER/.

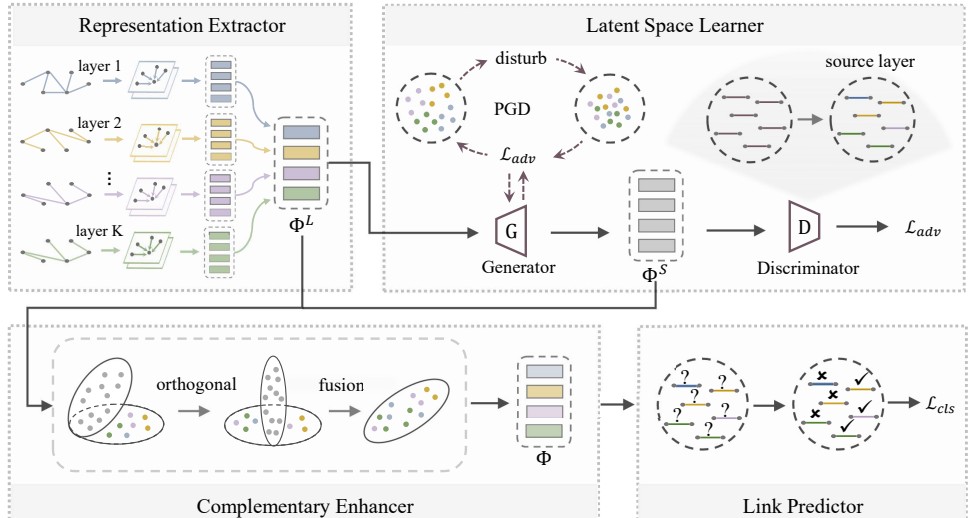

**Figure 2: Overview of the proposed model LUSTER. Representation Extractor learns the intra-layer representations of each layer. Latent Space Learner consists of a generator and a discriminator to obtain shared-latent space representations through adversarial training. Complementary Enhancer utilizes orthogonal fusion to combine intra-layer representations and shared-latent space representations. Link Predictor predicts whether a link is missing based on the enhanced representations.**

**Table 1: Notations**

| Symbol | Meaning | Symbol | Meaning |
|--------|---------|--------|---------|
| $\mathcal{G}$ | A multi-layer network | $\mathcal{G}_k$ | The $k$-th layer network |
| $\mathcal{V}$ | Observed nodes in $\mathcal{G}$ | $\mathcal{V}_k$ | Observed nodes in $\mathcal{G}_k$ |
| $\mathcal{E}$ | Observed links in $\mathcal{G}$ | $\mathcal{E}_k$ | Observed links in $\mathcal{G}_k$ |
| $\mathcal{E}^u$ | Unobserved links in $\mathcal{G}$ | $\mathcal{E}^u_k$ | Unobserved links in $\mathcal{G}_k$ |
| $\Phi^L$ | Intra-layer representations | $\phi^L_e$ | $\Phi^L$ for link $e$ |
| $\Phi^S$ | Shared-latent space representations | $\phi^S_e$ | $\Phi^S$ for link $e$ |
| $\Phi$ | Enhanced representations | $\phi_e$ | $\Phi$ for link $e$ |

more effectively [46]. In multi-modal learning [40], image processing [37], and natural language processing [3], it has been shown that effectively utilizing representations extracted from shared-latent space can improve overall model performance. However, this concept remains underutilized in link prediction within multi-layer networks. We aim to leverage a shared-latent space to mine inter-layer coupling and improve prediction accuracy.

### 2.3 Adversarial Neural Networks

Since the introduction of Generative Adversarial Networks (GANs) by Goodfellow et al. [17], the concept of adversarial training has gained widespread application. Notably, the Event Adversarial Neural Network proposed by Wang et al. [50] demonstrates effective transferable feature learning through adversarial training. We leverage this idea by integrating adversarial techniques into the construction of a shared-latent space within multi-layer networks. Through iterative adversarial training, we maintain cross-layer shared information, thereby enhancing the accuracy of link prediction.

More detailed discussions on related work are provided in Appendix A.1.

### 3 The Problem

Given a multi-layer network $\mathcal{G} = (\mathcal{V}, \mathcal{E})$, we aim to compute a set $\mathcal{P} = \{\langle e, \delta \rangle \mid e \in \mathcal{E}^u, \delta \in [0, 1]\}$, where for each unobserved link

$e \in \mathcal{E}^u$ is assigned a probability $\delta \in [0, 1]$ to quantify its existent likelihood. The perfect solution to this problem is that $\delta = 1$ for unobserved existent links and $\delta = 0$ for nonexistent links. We summarize the list of symbols used in this study in Table 1.

## 4 LUSTER

**Overview.** The workflow of LUSTER is shown in Fig. 2. It uses the representation extractor and the latent space learner to extract the intra-layer representations $\Phi^L$ and the shared-latent space representations $\Phi^S$. The latent space learner encompasses a generator and a discriminator. It uses a minimax two-player game, where the generator attempts to learn the shared-latent space representations to deceive the discriminator. While, the discriminator attempts to accurately distinguish the layer sources of links based on the representations learned by the generator. Then, the complementary enhancer combines the intra-layer representations $\Phi^L$ and the shared-latent space representations $\Phi^S$ via orthogonal fusion to obtain the enhanced representations $\Phi$. Finally, the link predictor is used on top of the complementary enhancer to predict missing links. Further details about the model components are as follows:

### 4.1 Representation Extractor

The representation extractor of LUSTER utilizes $K$ separate Graph Convolutional Networks (GCN) [22] to learn and/or extract the intra-layer representations of each individual layer. Specifically, for $k$-th layer, we obtain corresponding adjacency matrix $A_k$ and the initial matrix $H_k^{(0)}$ from the network graph $\mathcal{G}_k$. The convolution process for the $k$-layer may be denoted as:

$$H_k^{(l+1)} = \sigma(\tilde{D}_k^{-\frac{1}{2}} \tilde{A}_k \tilde{D}_k^{-\frac{1}{2}} H_k^{(l)} W_k^{(l)}) \tag{1}$$

where $\tilde{A}_k = A_k + I$ is the adjacency matrix $A_k$ with added self-loops, $\tilde{D}_k$ is the degree matrix of $\tilde{A}_k$, $W_k$ denotes the weight matrix,

and $\sigma(\cdot)$ is the ReLU activation function. In our case, we use a two-layered convolutional network. We use $N_k^L = H_k^{(2)} \in \mathbb{R}^{|\mathcal{V}| \times d_n}$ to denote the intra-layer representations of nodes in the $k$-th layer network, where $d_n$ denotes the dimension of the intra-layer representations of nodes. Subsequently, the intra-layer representations of all links in the $k$-th layer network is represented as:

$$\Phi_k^L = \{\phi_e^L | \phi_e^L = N_{k(e_l)}^L \oplus N_{k(e_r)}^L\}, \tag{2}$$

where $e_l$ and $e_r$ denote the left and right node of a link $e$ respectively, $\oplus$ denotes the concatenation operation, and $\phi_e^L \in \mathbb{R}^d$ denotes the intra-layer representation of the link $e$, with $d = 2d_n$. We use $\Phi^L = \{\Phi_k^L\}_{k=1}^K \in \mathbb{R}^{|\mathcal{E} \cup \mathcal{E}^u| \times d}$ to denote the intra-layer representations of links in all the layers.

We denote the representation extractor as $M_L(\mathcal{G}; \theta_L)$, where $\mathcal{G}$ denotes the original multi-layer network and $\theta_L$ denotes all parameters in the representation extractor.

## 4.2 Latent Space Learner

The latent space learner of LUSTER uses a generator-discriminator architecture, where the objective of the generator is to compute the shared-latent space representations of links across different layers. At the same time, the discriminator attempts to improve the ability of the generator by effectively distinguishing the differences between the link representations provided by the generator from different layers. Further details are as follows:

**Generator.** To learn the shared-latent space representations across different layers, the generator uses Convolutional Neural Network (CNN) to learn from the intra-layer representations obtained by the representation extractor. We argue CNN can effectively integrate information across different layers thus computing shared-latent space representations indicative of interconnections among different layers. The convolution operation of the $h$ consecutive links, starting from link $e$, can be mathematically expressed as:

$$\phi_e^S = \sigma(\sum_{i=0}^{h-1} W_i \cdot \phi_{e+i}^L), \tag{3}$$

where $\sigma(\cdot)$ denotes the ReLU activation function and $W_i$ denotes the weight of the convolution filter. $\Phi^S = \{\phi_e^S | e \in \mathcal{E} \cup \mathcal{E}^u\} \in \mathbb{R}^{|\mathcal{E} \cup \mathcal{E}^u| \times d}$ denotes the shared-latent space representations of links in the multi-layer network. The dimension of shared-latent space representations is same as that of intra-layer representations.

**Discriminator.** Specifically, for a given link sample $e$, the purpose of the discriminator is to distinguish which layer of the multi-layer network the link $e$ originates from. In our case, the latent space learner uses a discriminator consisting of a fully connected layer with softmax activation function, as shown below:

$$d_e = \text{softmax}(W^T \cdot \phi_e^S + b), \tag{4}$$

where $W \in \mathbb{R}^{d \times K}$ and $b \in \mathbb{R}^{K \times 1}$ denote the weight matrix and the bias vector of the fully connected layer, respectively. $W^T$ denotes the transpose of $W$ and $d_e \in \mathbb{R}^{K \times 1}$ denotes the probability that the link originates from each layer. We use cross-entropy loss as the loss of the discriminator, shown as follows:

$$\mathcal{L}_{adv} = -[\mathbb{E}_{e \sim \mathcal{E}} \sum_{k=1}^K y_{ek} log(d_{ek}) + \mathbb{E}_{e \sim \mathcal{E}^u} \sum_{k=1}^K y_{ek} log(d_{ek})], \tag{5}$$

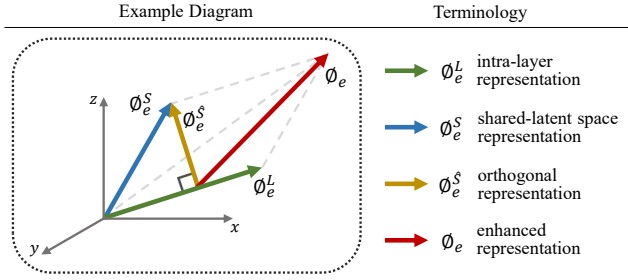

| Example Diagram | Terminology |
|---|---|
| | $\emptyset_e^L$   intra-layer representation |
| | $\emptyset_e^S$   shared-latent space representation |
| | $\emptyset_e^{\hat{S}}$   orthogonal representation |
| | $\emptyset_e$   enhanced representation |

**Figure 3: Fusion of the intra-layer representation $\phi_e^L$ and shared-latent space representation $\phi_e^S$ for link $e$.**

where $y_{ek}$ indicates the ground truth label: 1 if link $e$ belongs to the $k$-th layer, and 0 otherwise. $d_{ek}$ represents the predicted probability by the discriminator for link $e$ belonging to the $k$-th layer.

A lower loss indicates that the $\Phi^S$ helps the discriminator to distinguish different layers more effectively, while a larger loss reflects that the $\Phi^S$ given by the generator can deceive the discriminator. For this, a minimax game is established between the generator and the discriminator, where on one hand, the generator continuously learns shared-latent space representations to deceive the discriminator and strives to maximize $\mathcal{L}_{adv}$. While, on other hand, in order to avoid being deceived, the discriminator aims to minimize $\mathcal{L}_{adv}$.

We use $M_S(\Phi^L; \theta_G, \theta_D)$ to denote the latent space learner, where $\theta_G$ and $\theta_D$ denote all parameter that the generator and the discriminator need to learn, respectively.

## 4.3 Complementary Enhancer

The complementary enhancer utilizes orthogonal fusion to integrate intra-layer representations and shared-latent space representations. Since the shared-latent space representations are derived from multiple intra-layer representations, they may incur linear dependencies. By removing the overlapping components and retaining the orthogonal parts, we aim to acquire a more effective linear combination. For this, we apply orthogonal projection between the intra-layer representations and shared-latent space representations to extract complementary components, later combine them with the original intra-layer representations.

In this aspect of projection, we follow existing work by Qin et al. [39] that proposed orthogonal projection layer (OPL) in order to map traditional features into a semantic space orthogonal to common features, yielding "pure representations" in order to improve the classification performance. Specifically, for link $e \in \mathcal{E} \cup \mathcal{E}^u$, the projection of shared-latent space representation $\phi_e^S$ in a direction orthogonal to intra-layer representation $\phi_e^L$ is expressed as:

$$\phi_e^{\hat{S}} = \text{ortho}\langle \phi_e^S, \phi_e^L \rangle = \frac{\|\phi_e^S \times \phi_e^L\|_2}{\|\phi_e^S\|_2 \cdot \|\phi_e^L\|_2} \cdot \phi_e^S \tag{6}$$

where $\| \cdot \|_2$ refers to the $\mathcal{L}_2$ norm operator. Then, the complementary enhancer combines the intra-layer representation $\phi_e^L$ with the orthogonal representation $\phi_e^{\hat{S}}$ of link $e$, as follows:

$$\phi_e = \phi_e^L[i] + \phi_e^{\hat{S}}[i], i = 1, 2, \cdots, d, \tag{7}$$

where $\phi_e$ denotes the enhanced representation of link $e$, $\Phi = \{\phi_e | e \in \mathcal{E} \cup \mathcal{E}^u\} \in \mathbb{R}^{|\mathcal{E} \cup \mathcal{E}^u| \times d}$ denotes the enhanced representations of all the links in the multi-layer network.

For a specific link $e$, the process of combining intra-layer representation and shared-latent space representation is illustrated in Fig. 3. It is evident from Fig. 3 that the enhanced representation $\phi_e$ obtained by Eq. 7 is equivalent to $\phi_e^L + \phi_e^S - \mathrm{proj}\langle \phi_e^S, \phi_e^L\rangle$, where $\mathrm{proj}\langle \phi_e^S, \phi_e^L\rangle = \|\phi_e^S\|_2 \cdot cos\langle \phi_e^S, \phi_e^L\rangle \cdot \phi_e^L$ represents the projection of $\phi_e^S$ in the direction of $\phi_e^L$, $cos\langle \phi_e^S, \phi_e^L\rangle = \frac{\phi_e^S \cdot \phi_e^L}{\|\phi_e^S\|_2 \cdot \|\phi_e^L\|_2}$. The $\mathrm{proj}\langle \phi_e^S, \phi_e^L\rangle$ represents the overlapping components between the intra-layer representation $\phi_e^L$ and shared-latent space representation $\phi_e^S$. Through the above orthogonal fusion, the complementary enhancer successfully removes the overlapping components $\mathrm{proj}\langle \phi_e^S, \phi_e^L\rangle$ while retaining the orthogonal parts $\phi_e^{\widehat{S}}$. This results in a more reasonable linear combination for the link, thus yielding the enhanced representation $\phi_e$.

We use $M_E(\Phi^L, \Phi^S; \theta_E)$ to denote the complementary enhancer, where $\theta_E$ denotes all parameters in the module. The representation extractor and the latent space learner pass $\Phi^L = \{\phi_e^L | e \in \mathcal{E} \cup \mathcal{E}^u\}$ and $\Phi^S = \{\phi_e^S | e \in \mathcal{E} \cup \mathcal{E}^u\}$ to the complementary enhancer.

## 4.4 Link Predictor

The link predictor is built on top of the complementary enhancer. It uses the enhanced representations $\Phi = \{\phi_e | e \in \mathcal{E} \cup \mathcal{E}^u\} \in \mathbb{R}^{|\mathcal{E} \cup \mathcal{E}^u| \times d}$ as input and passes them through a fully connected layer in order to compute the prediction results. Formally, for a given link sample $e$, we compute the probability of the existence of link, *i.e.*, $p_e$ as follows:

$$p_e = \mathrm{softmax}(W_p^T \cdot \phi_e + b_p), \tag{8}$$

where $W_p \in \mathbb{R}^{d \times 2}$ and $b_p \in \mathbb{R}^{2 \times 1}$ denote the weight matrix and the bias vector respectively. We use the cross-entropy function as the prediction loss, defined as follows:

$$\mathcal{L}_{cls} = -[\mathop{\mathbb{E}}_{e \sim \mathcal{E}} log(p_e) + \mathop{\mathbb{E}}_{e \sim \mathcal{E}^u} log(1 - p_e)]. \tag{9}$$

We use $M_P(\Phi; \theta_P)$ to denote the link predictor, where $\theta_P$ represents the parameters of the predictor.

## 4.5 Model Integration

We define the overall loss function of LUSTER, as follows:

$$\mathcal{L} = \mathcal{L}_{cls} + \mathcal{L}_{adv}. \tag{10}$$

For model training, we employ the Adam algorithm [21] and decay learning rate [15] for model optimization. The process of updating parameters is as follows:

$$\theta^{(t+1)} = \theta^{(t)} - \eta_r(\nabla_\theta \mathcal{L}), \tag{11}$$

where $\theta = \{\theta_L, \theta_D, \theta_E, \theta_P\}$ and $\eta_r$ denotes the learning rate, which decays with epochs during the training stage:

$$\eta_r = \frac{\eta_0}{(1 + \alpha \times r)^\beta}, \tag{12}$$

where $r$ denotes the ratio of the current epoch to the total number of epochs, $\eta_0 = 0.01$ denotes the initial learning rate. $\alpha = 10$ and $\beta = 0.75$ are hyperparameters, which are the same as those of [15].

During the minimax two-player game between the generator and the discriminator, the discriminator attempts to update the parameters $\theta_G$ in the direction of gradient descent to minimize $\mathcal{L}_{adv}$, while the generator continuously disturbs the parameters $\theta_G$ in the direction of gradient ascent to maximize $\mathcal{L}_{adv}$. This dynamic process helps the model to better capture the shared-latent space representations across different layers.

Specifically, to implement the adversarial process, a common basic operation is to introduce a gradient reversal layer (GRL) [49] that inverts the gradient of $\theta_G$ during back-propagation. However, the loss $\mathcal{L}_{adv}$ is usually not linear with respect to the parameters $\theta_G$, meaning that there may be some disturbance that can cause the gradient increase more relative to a direct inverse. For this, we introduce the projected gradient descent (PGD) [28] with minor adjustments. The original PGD affects the parameters of the generator and its preceding modules. We focus on disturbing only the gradients of the generator during the optimization process, avoiding interference with the preceding module (*i.e.*, the representation learner) and preventing any negative impact on the extraction of inter-layer representations. During the back-propagation process, the parameters $\theta_G$ contained in the generator are disturbed $N$ times. The cumulative disturbance up to $n = \{1, 2, \cdots, N\}$ times is:

$$c^{(n)} = \sum_{i=0}^{n-1} [\mu \cdot sgn(\nabla_{\theta_G^{(i)}} \mathcal{L}_{adv}^{(i)})], \tag{13}$$

where $\mu$ denotes the disturbance coefficient and $sgn(\cdot)$ denotes sign function. $\theta_G^{(0)}$ and $\mathcal{L}_{adv}^{(0)}$ denote the original states prior to the onset of disturbances. $\theta_G^{(n)}$ and $\mathcal{L}_{adv}^{(n)}$ denote the states after experiencing disturbances up to $n = \{1, 2, \cdots, N\}$ times, respectively. After each disturbance, if the cumulative disturbance $c^{(n)}$ exceeds the space of disturbance radius $\lambda$, it is projected back onto the spherical surface with a radius of $\lambda$ to ensure that the disturbance is not too large. The process can be formulated as:

$$\theta_G^{(n)} = \begin{cases} \theta_G^{(0)} + \frac{\lambda}{\|c^{(n)}\|_2} c^{(n)}, & \text{if } \|c^{(n)}\|_2 > \lambda \\ \theta_G^{(0)} + c^{(n)}, & otherwise \end{cases} \tag{14}$$

where $\|\cdot\|_2$ refers to the $\mathcal{L}_2$ norm operator. During each disturbance, the discrimination loss $\mathcal{L}_{adv}^{(n)}$ is calculated according to Eq.5 after obtaining $\theta_G^{(n)}$. After $N$ disturbances, we update the generator parameters $\theta_G$ as follows:

$$\theta_G^{(t+1)} = \theta_G^{(t)} - \eta_r(\nabla_{\theta_G} \mathcal{L} + \nabla_{\theta_G^{(N)}} \mathcal{L}_{adv}^{(N)}). \tag{15}$$

Here, we not only consider the gradient of the current loss $\mathcal{L}$ with respect to the generator parameters $\theta_G$, we also incorporate the gradient of the loss $\mathcal{L}_{adv}^{(N)}$ with respect to $\theta_G^{(N)}$. We claim this update process combines the effects of both the current loss and the loss after $N$ disturbances, thereby introducing disturbance information into the parameter updates in order to compute the shared-latent space representations.

**Training Workflow.** The detailed training steps of our proposed LUSTER are summarized in Algorithm 1, and explained below. Initially, LUSTER takes the multi-layer network graph $\mathcal{G}$, and initial learning rate $\eta_0$ as inputs. It then extract the actual layer labels of links, *i.e.,* layers corresponding to links, and prediction labels

**Algorithm 1:** LUSTER Training Workflow

**Input:** A multi-layer network graph $\mathcal{G}$ and the initial learning rate $\eta_0$.

**Output:** Predict each unobserved link $e$ in $\mathcal{E}^u$

1   Obtain the actual layer labels of links;

2   Obtain the actual prediction labels of links;

3   **for** each epoch **do**

4      Update the learning rate $\eta_r$ by Eq.12;

5      Obtain $\Phi^L$, $\Phi^S$ and $\Phi$ by Eq.2, Eq.3 and Eq.7;

6      Calculate $\mathcal{L}_{adv}$, $\mathcal{L}_{cls}$ and $\mathcal{L}$ by Eq.5, Eq.9 and Eq.10;

7      **for** each disturbance **do**

8         Calculate cumulative disturbance $c^{(t)}$ by Eq.13;

9         Calculate parameter $\theta_G^{(t)}$ by Eq.14;

10        Calculate loss $\mathcal{L}_{adv}^{(t)}$ by Eq.5;

11      **end**

12      Update parameters $\theta_L, \theta_D, \theta_E, \theta_P$ by Eq.11;

13      Update parameter $\theta_G$ by Eq.15;

14   **end**

15   **for** each unobserved link $e$ in $\mathcal{E}^u$ **do**

16      Calculate $p_e$ by Eq.8;

17   **end**

of links, *i.e.,* whether the links exist or not (lines 1-2). For each training epoch, it recomputes the learning rate as $\eta_r$ using Eq. 12 (line-4). Subsequently, the intra-layer representations $\Phi^L$ are obtained through the representation extractor; the shared-latent space representations $\Phi^S$ is obtained through the latent space learner; and the enhanced representations $\Phi$ are obtained through the complementary enhancer (line 5). Next, $\mathcal{L}_{adv}$, $\mathcal{L}_{cls}$ and $\mathcal{L}$ are calculated (line 6). Later, the gradient associated with the generator is disturbed multiple times (lines 8-10), and the model parameters are updated (lines 12-13). Finally, the probability of existence of each unobserved link is predicted (line 16).

## 5 Experimentation

In this Section, we perform a rigorous experimental evaluation of LUSTER using benchmark datasets compared against existing state-of-the-art methods as baselines.

### 5.1 Experimental settings

**Datasets.** To fairly evaluate the performance of the proposed LUSTER, we consider the following real-world multi-layer networks: (i) Aarhus [29]; (ii) Enron [45]; (iii) Kapferer [11]; (iv) LonRail [11]; (v) TF [18]; and (vi) Reddit [24]. The statistics of these multi-layer network datasets are shown in Table 2. Detailed description of these datasets are given in Appendix A.2.

**Baselines.** To demonstrate the effectiveness of LUSTER, we compare it with the following existing state-of-the-art methods: (i) Adamic Adar [2]; (ii) Jaccard [43]; (iii) NSILR [54]; (iv) SEAL [55]; (v) MultiSup [41]; (vi) MADM [27]; (vii) MANE [6]; (viii) MNERLP [32]; and (ix) HOPLP [33]. Further details about these baseline models are provided in the Appendix A.3.

**Evaluation Metrics.** For performance evaluation, we use Accuracy (Acc) and Area under the ROC Curve (AUC) as our evaluation

**Table 2: Statistics of several multi-layer network datasets.**

| Datasets | #Nodes | #Edges | $k$ | $|\mathcal{V}_k|$ | $|\mathcal{E}_k|$ |
|---|---|---|---|---|---|
| Aarhus [29] | 61 | 620 | 1 | 60 | 193 |
| | | | 2 | 32 | 124 |
| | | | 3 | 25 | 21 |
| | | | 4 | 47 | 88 |
| | | | 5 | 60 | 194 |
| Enron [45] | 151 | 261 | 1 | 142 | 133 |
| | | | 2 | 117 | 128 |
| Kapferer [11] | 39 | 552 | 1 | 39 | 158 |
| | | | 2 | 39 | 223 |
| | | | 3 | 35 | 76 |
| | | | 4 | 37 | 95 |
| LonRail [11] | 369 | 441 | 1 | 271 | 312 |
| | | | 2 | 83 | 83 |
| | | | 3 | 45 | 46 |
| TF [18] | 1564 | 32579 | 1 | 1564 | 14090 |
| | | | 2 | 1508 | 18471 |
| Reddit [24] | 67180 | 858488 | 1 | 54075 | 571927 |
| | | | 2 | 35776 | 286561 |

metrics. Detailed descriptions and mathematical formulations of these metrics are given in Appendix A.4.

**Experimental Setup.** In evaluating model performance on these datasets, we adopt a standard approach where observed links are considered positive samples, and unobserved links are treated as negative samples. The data is randomly split into training, validation, and testing sets in ratio of: 8:1:1 to ensure robust evaluation. For representation extractor, we maintain a consistent hidden layer dimension of 64 for all GCNs. The dimension $d_n$ of the node intra-layer representations $N_k^L \in \mathbb{R}^{|\mathcal{V}| \times d_n}$ is set to 16. Consequently, the dimensions of $\Phi^L \in \mathbb{R}^{|\mathcal{E} \cup \mathcal{E}^u| \times d}$, $\Phi^S \in \mathbb{R}^{|\mathcal{E} \cup \mathcal{E}^u| \times d}$, and $\Phi \in \mathbb{R}^{|\mathcal{E} \cup \mathcal{E}^u| \times d}$ are all configured to be $d = 2d_n = 32$. For adversarial training, we set the disturbance coefficient to $\mu = 3$ and the disturbance radius to $\lambda = 7$, applying a total of $N = 4$ disturbances. Batch size is set to 128, and the training is conducted for a maximum of 1000 epochs with early stopping [38]. Our framework is implemented in Python 3.8 and PyTorch 2.1 on an NVIDIA RTX A100 GPU. For baseline methods, we follow their respective papers and fine-tune models based on recommended parameter settings.

### 5.2 Main Results

Table 3 shows the results of LUSTER. Here, we report the Acc and AUC for the proposed model compared against different baseline models using different evaluation benchmarks. It is evident that the proposed model, *i.e.,* LUSTER, outperforms the baseline models across both metrics, verifying the effectiveness of our method. For instance, for Aarhus dataset, it improves the Acc and AUC scores by 4.13% and 15.87% respectively compared to the second best.

We attribute this performance improvement to multiple different factors, enumerated as follows: *Firstly*, LUSTER chooses to retain the multi-layer network structure rather than merging it into a weighted single-layer network. This approach maximally preserves multi-layer structural information, allowing for the accurate acquisition of intra-layer representations while avoiding potential information loss during the conversion to a single-layer network. *Secondly*, LUSTER extracts representations from a shared-latent

**Table 3: Prediction results of several models on real-world datasets in terms of Acc(%) and AUC(%). The boldface scores indicate the best results, while the underlined scores indicate the second-best results.**

| Datasets | Metrics | Adamic Adar [2] | Jaccard [43] | NSILR [54] | SEAL [55] | MultiSup [41] | MADM [27] | MANE [6] | MNERLP [32] | HOPLP [33] | LUSTER (%-Improv.) |
|---|---|---|---|---|---|---|---|---|---|---|---|
| Aarhus [29] | Acc | 75.65±0.29 | 72.46±0.18 | 76.45±1.35 | 72.86±1.17 | 75.61±1.85 | 71.02±1.26 | 59.79±1.30 | 81.82±1.67 | 81.47±1.83 | **85.20±1.40** (4.13%) |
| | AUC | 64.77±0.40 | 64.52±0.43 | 78.40±2.63 | 76.49±2.20 | 75.65±2.86 | 71.31±1.77 | 62.84±1.66 | 77.98±2.05 | 76.64±2.25 | **90.84±1.05** (15.87%) |
| Enron [45] | Acc | 56.67±0.82 | 56.60±0.69 | 74.43±1.11 | 67.33±0.94 | 71.16±0.64 | 50.38±2.09 | 70.57±1.21 | 73.12±0.54 | 73.46±0.41 | **86.12±1.95** (15.71%) |
| | AUC | 56.77±0.82 | 56.49±0.69 | 60.48±1.11 | 69.28±1.59 | 63.98±0.85 | 50.96±1.96 | 81.10±0.51 | 58.86±0.54 | 59.06±0.42 | **93.26±1.78** (14.99%) |
| Kapferer [11] | Acc | 64.58±0.23 | 58.25±0.09 | 68.98±1.53 | 69.76±1.34 | 63.83±1.23 | 73.53±1.42 | 54.23±1.44 | 75.51±1.46 | 69.88±1.48 | **82.17±1.63** (8.82%) |
| | AUC | 58.25±0.35 | 57.24±0.39 | 72.99±2.59 | 72.37±0.12 | 63.65±0.40 | 80.36±1.69 | 56.46±1.53 | 73.06±1.22 | 71.43±2.96 | **86.28±2.95** (7.37%) |
| LonRail [11] | Acc | 51.81±0.36 | 51.42±0.13 | 67.35±0.37 | 75.58±0.13 | 80.93±1.48 | 55.66±1.01 | 66.67±1.34 | 68.88±1.49 | 75.38±1.52 | **89.54±1.21** (10.64%) |
| | AUC | 51.81±0.36 | 51.42±0.13 | 60.14±0.37 | 84.34±1.06 | 80.35±1.19 | 55.78±1.01 | 79.23±0.79 | 61.33±1.49 | 61.04±1.51 | **92.03±2.93** (9.12%) |
| TF [18] | Acc | 75.95±0.29 | 50.38±0.70 | 83.72±1.35 | 86.14±0.21 | 73.82±0.25 | 73.67±1.53 | 52.17±0.40 | 85.17±2.09 | 86.05±1.02 | **91.67±1.20** (6.42%) |
| | AUC | 84.06±0.47 | 83.09±0.38 | 80.54±2.10 | 86.23±0.20 | 75.73±1.82 | 74.73±1.42 | 58.17±1.64 | 85.32±2.93 | 85.46±2.59 | **89.31±0.61** (3.57%) |
| Reddit [24] | Acc | 79.93±0.16 | 50.15±0.13 | 73.65±0.46 | 88.17±0.39 | 73.38±0.36 | 50.18±0.06 | 66.42±0.21 | 74.07±1.18 | 73.93±1.12 | **89.10±0.63** (1.05%) |
| | AUC | 86.75±0.08 | 85.29±0.07 | 88.52±0.49 | 93.06±0.34 | 80.10±0.76 | 86.27±0.59 | 81.45±0.20 | 87.98±0.29 | 88.19±0.53 | **96.02±0.53** (3.18%) |

space across different layers through adversarial training. This process enhances the interaction of information between layers and further improves the understanding of inter-layer coupling in multi-layer networks. *Thirdly*, LUSTER introduces an orthogonal fusion strategy that effectively combines the intra-layer representations with the shared-latent space representations. This fusion method preserves the unique features of each layer while minimizing redundancy, thereby increasing the efficiency of shared information utilization and ultimately enhancing the overall performance.

Comparing the results amongst the baseline models, we observe SEAL and MNERLP demonstrate comparatively better performance than other baseline models across several datasets. Specifically, SEAL effectively avoids the interference of irrelevant information by extracting local subgraphs, while introducing layer information to enable the model to capture the relationship and features between different layers. On the other hand, MNERLP combines local and global representations to calculate the node and edge relevance, thereby achieving more effective link prediction. By comprehensively considering these factors, these models demonstrate relatively good prediction capabilities.

## 5.3 Ablation Study

We investigate the impact of various model components of the proposed model, *i.e.,* LUSTER on link prediction. For this, we ablate different model components in order to understand the contribution of each individual model component. Specifically, we propose following different variants of LUSTER:

(i) *w/o* Latent Space Learner (−S): In this variant, we replace the latent space learner module with a fully connected layer while keeping the remaining modules unchanged. This change allows us to assess the significance of using adversarial training between the generator and discriminator for extracting representations from a shared-latent space across different layers..

(ii) *w/o* Complementary Enhancer (−E): Here, we replace the complementary enhancer module with a simple element-wise addition operation and leave the remaining modules unchanged. This comparison helps evaluate the effectiveness of using orthogonal fusion technique when integrating two different representations.

(iii) *w/o* Latent Space Learner and Complementary Enhancer (−S&E): This variant involves the removal of both the latent space learner

**Table 4: Ablation study on several datasets in terms of Acc (%) and AUC (%). Boldface scores indicate the best results.**

| Datasets | Metrics | LUSTER | −S | −E | −S&E |
|---|---|---|---|---|---|
| Aarhus [29] | Acc | **85.20** | 84.28 | 82.67 | 81.63 |
| | AUC | **90.84** | 80.87 | 77.59 | 69.84 |
| Enron [45] | Acc | **86.12** | 83.39 | 81.30 | 80.29 |
| | AUC | **93.26** | 89.61 | 86.54 | 83.55 |
| Kapferer [11] | Acc | **82.17** | 80.54 | 74.95 | 72.67 |
| | AUC | **86.28** | 78.62 | 74.05 | 68.17 |
| LonRail [11] | Acc | **89.54** | 87.48 | 86.50 | 68.10 |
| | AUC | **92.03** | 82.83 | 68.42 | 66.92 |
| TF [18] | Acc | **91.67** | 88.31 | 87.45 | 85.73 |
| | AUC | **89.31** | 87.29 | 85.60 | 84.39 |
| Reddit [24] | Acc | **89.10** | 87.98 | 85.69 | 84.10 |
| | AUC | **96.02** | 95.12 | 93.28 | 90.51 |

and the complementary enhancer modules, while the remaining modules are left unchanged. This comparison allows us to analyze the complementary impact of excluding both components on the overall performance of LUSTER.

We analyze the results of the ablation study from different perspectives, with quantitative analysis presented in Section 5.3.1, and qualitative analysis presented in Section 5.3.2.

*5.3.1 Quantitative Analysis.* We evaluate the performance of LUSTER and its three variants across several real-world datasets, in terms of Acc and AUC as evaluation metrics, with results presented in Table 4. The results indicate a significant decline in model performance for the ablation variants, underscoring the importance of individual components. By comprehensively examining the data in Table 4, we draw the following conclusions:

**(i) Individual Components.** Comparing the results of ablating individual model components, (*i.e.,* −S, −E), we observe that overall both ablation variants result in a decrease in the model performance. This is explained by the fact that on one hand, the latent space learner utilizes adversarial training to learn robust representations that are shared across different layers of the multi-layer networks in order to ensure that LUSTER captures common features and structural correlations essential for the prediction task. On the other hand, the complementary enhancer introduces the

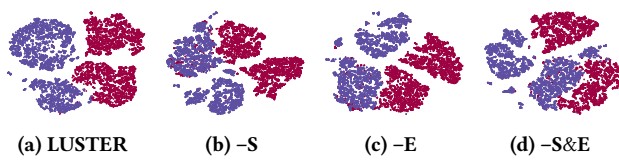

(a) LUSTER        (b) −S        (c) −E        (d) −S&E

**Figure 4: T-SNE visualizations of the representations before the link predictor that are learned by LUSTER and its three variants on the TF dataset.**

orthogonal fusion technique to seamlessly combine intra-layer representations and shared-latent space representations. This approach prevents redundancy and enhances predictive capabilities of model by leveraging diverse aspects of the network structure.

**(ii) Multiple Components.** We observe that the removing both the latent space learner and the complementary enhancer (−S&E) exhibit a complementary effect in performance reduction. For instance, compared to the complete model, −S&E experiences a decrease of 23.12%, 10.41%, 20.99%, 27.28%, 5.51%, and 5.74% for the Aarhus, Enron, Kapferer, LonRail, TF, and Reddit datasets, respectively for the AUC metric. We argue that by excluding both components, the model fails to capture essential structural correlations and comprehensive network representations, resulting in significantly compromised performance in link prediction tasks. This underscores the complementary roles of both model components in maximizing effectiveness of LUSTER.

In conclusion, LUSTER synergistically integrates the latent space learner and the complementary enhancer to leverage their complementary strengths. This integration enhances the ability of LUSTER to capture and utilize the shared-latent space representations between layers in multi-layer networks, thereby significantly improving prediction performance across diverse datasets.

*5.3.2 Qualitative Analysis.* To qualitatively analyze the effectiveness of LUSTER, we use t-SNE [48] to visualize the representations before the link predictor that are learned by LUSTER and its three variants on the TF dataset. From Fig. 4, we observe a clear distinctive boundary between different label clusters in LUSTER compared to its variants. This demarcation indicates that the representations learned by LUSTER are more separable and informative. Such enhanced discriminative ability suggests that LUSTER effectively captures and utilizes meaningful features for link prediction tasks, thereby demonstrating its efficacy.

Overall, this ablation study reinforces the effectiveness of individual components of LUSTER. By leveraging several techniques like adversarial training and orthogonal fusion, LUSTER enhances the predictive power of multi-layer network analysis. This holistic approach ensures that LUSTER not only learns robust representations but also integrates them effectively to improve link prediction accuracy across diverse real-world datasets.

## 5.4 Case Study

We observe that active nodes have a significant influence within network structures, such as active users in social networks, prolific authors in academic collaboration networks, and frequent traders on e-commerce platforms. The connections between these nodes have a greater impact on the overall functionality and structure of the network. Therefore, focusing on link prediction analysis among

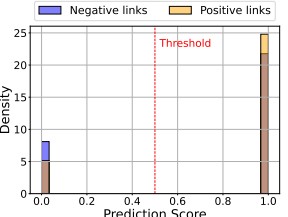
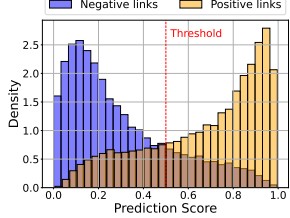

(a) Randomly Initialized Model        (b) After Training

**Figure 5: Prediction results of the active subgraph in the Reddit dataset.**

active nodes allows for a better understanding of the potential relationships between these key nodes, bringing significant effects and value in practical applications. To comprehend the performance of LUSTER for high-impact/active nodes, we conduct a case study. For this, we use Reddit dataset, and select nodes that have a significant impact on the network structure, *i.e.,* active nodes with degrees greater than or equal to 5 in both layers. Based on these nodes, we construct an active two-layer subgraph. Since these nodes have high connectivity in both layers, with more neighbors and richer connection information, they are more likely to exhibit similar behavioral patterns or share information across layers. This coupling makes these nodes an ideal foundation for learning representations in the shared-latent space, thereby enabling more effective capture of cross-layer commonalities.

We plot the prediction score distribution in Fig. 5. Fig. 5a illustrates the results of the randomly initialized model before training, which exhibits extreme randomness, with nearly all link prediction scores close to 0 or 1. At the classification threshold of 0.5, most classification results are inaccurate, indicating that there is no clear distinction between positive and negative samples at the initial stage. In contrast, Fig. 5b shows the prediction results of the model after training, indicating that negative links are primarily concentrated in the lower score range with prediction scores generally approaching 0, while positive links are predominantly found in the higher score range, with prediction scores approaching 1. Compared to Fig. 5a, the distribution in Fig. 5b demonstrates the excellent performance of LUSTER in distinguishing between positive and negative links. This indicates that LUSTER effectively captures the key features of links within the active subgraph, showcasing its ability to accurately identify potential relationships among high-impact nodes. Overall, the case study demonstrates that LUSTER is not only effective within the overall network but also further confirms its predictive performance concerning core nodes and their interrelationships, providing significant value for practical applications such as recommendation systems and collaborative networks by enhancing recommendation quality and user experience.

## 6 Conclusion and Future Work

In this research, we propose a novel framework named **L**ink prediction **U**tilizing **S**hared-la**T**ent spac**E R**epresentation (LUSTER) in multi-layer networks. Comprehensive experimental evaluation demonstrates that LUSTER outperforms the baseline models by a significant margin. In future, we aim to extend this work to dynamic temporal networks.

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

# A  Appendix

## A.1  Supplement to Related Work

*A.1.1  Shared-Latent Space.* Shared-latent space is a mechanism for bridging information between multiple modalities and has attracted much attention in many research fields in recent years [14]. It refers to a unified feature space that can integrate information from different data sources in order to more effectively capture potential relationships [46]. In fields such as multi-modal learning [40], image processing [37], and natural language processing [3], researchers have found that different types of data can often complement each other through a shared-latent space, thereby improving the overall performance of the model. The significance of building a shared-latent space is that it can help the model overcome the challenges brought by differences between modalities and data missing [40]. For example, in multi-modal sentiment analysis [36], data from different modalities such as text, audio, and video can work together in a shared-latent space to more fully understand emotional expressions. In the study of graph neural networks [7], building a shared-latent space also enables the attributes of different nodes to be effectively fused, improving the performance of node classification and link prediction.

At present, many researchers have proposed a variety of methods to build and utilize a shared-latent space to improve the performance of models when processing complex data. Structural Attribute Transformer (SAT) proposed by Chen et al. [7] assumes that there is a shared-latent space between graph structure and node attributes, decouples the two through distribution matching technology, successfully handles link prediction and node attribute completion tasks, and achieves excellent performance on graph datasets with missing attributes. Graph Complete Network (GC-Net) by Lian et al. [25] optimizes complete and incomplete multimodal data in a shared-latent space to address the problem of incomplete modality in conversations, combining "speaker graph neural network" and "temporal graph neural network", and demonstrates superior performance on multimodal conversation datasets. Causality-Invariant Interactive Mining (CIIM) proposed by Yan et al. [53] eliminates modality bias through causal intervention and learns modality-consistent feature embedding in a shared-latent space. Experimental results show its superiority on multiple cross-modal tasks. However, this concept has not been fully applied in link prediction in multi-layer networks. Therefore, our research aims to extract representations from a shared-latent space to effectively mine cross-layer shared information and further improve prediction performance. Through this exploration, we hope to open up new directions for the analysis and application of multi-layer networks.

*A.1.2  Adversarial Neural Networks.* After the seminal proposal of Generative Adversarial Networks (GANs) by Goodfellow et al. [17], the concept of adversarial training gained immense popularity and has been successfully applied across various domains, including domain adaptation [16], semi-supervised classification [4], fake news detection [50], anomaly detection [5], etc. Among the mentioned work, the Event Adversarial Neural Network (EANN), proposed by Wang et al. [50] has garnered attention for its effectiveness in transferable feature learning. EANN employs adversarial training between the multi-modal feature extractor and event discriminator to effectively identify and retain common features that can transfer across different events, while discarding event-specific features that cannot transfer. This approach has proven particularly useful for detecting fake news in emerging events, significantly improving the adaptability of the model in handling diverse event data. Inspired by the success of transferable feature learning in event detection, we pioneer the integration of adversarial techniques into the construction of a shared-latent space in multi-layer networks, addressing a major gap in this field. Through iterative adversarial training between the generator and discriminator, our model reduces its dependence on any specific layer while preserving inter-layer coupling, leading to improved link prediction accuracy. By constructing a shared-latent space using adversarial training, the adaptability to complex structures of multi-layer networks is enhanced.

## A.2  Datasets

**(i) Aarhus** [29] represents a 5-layer network formed among the employees of the Aarhus computer science department. These five layers represent different types of relationships among the employees, including Facebook connections, leisure activities, work-related interactions, collaborative writing, and lunch interactions.

**(ii) Enron** [45] represents a 2-layer network between employees, denoting their relationships with superiors and colleagues, respectively.

**(iii) Kapferer** [11] represents a 4-layer network observed in a tailor shop over a period of ten months, depicting work, assistance, friendship, and emotional relationships, respectively.

**(iv) LonRail** [11] represents a 3-layer network that represents railway stations in London. The network comprises three layers, denoting stations connected by the underground, above ground, and DLR (Docklands Light Railway), respectively.

**(v) TF** [18] represents a 2-layer network formed between Twitter and Foursquare. The first layer represents follow relationships on Twitter, and the second layer represents friendship relationships on Foursquare.

**(vi) Reddit** [24] represents a 2-layer network extracted from posts with hyperlinks between subreddits. The first layer captures hyperlinks in post titles, while the second captures those in post bodies, each reflecting distinct forms of subreddit interactions.

These datasets encompass a wide range of network types and relationships, making them a robust benchmark for evaluating the performance of LUSTER. Their diverse structures and applications in real-world scenarios enable a comprehensive assessment of the effectiveness and applicability of LUSTER across various domains.

## A.3  Baselines

**(i) Adamic Adar** [2] uses the Adamic-Adar coefficient to measure the similarity between two nodes after transforming a multi-layer network into a weighted single-layer network.

**(ii) Jaccard** [43] employs the Jaccard coefficient to measure the similarity between nodes after transforming a multi-layer network into a weighted single-layer network.

**(iii) NSILR** [54] proposes a node similarity index based on layer relevance of the multi-layer network by utilizing intra-layer and inter-layer representations.

**(iv) SEAL** [55] first constructs adjacency subgraphs, and then uses DGCNN to learn features of these subgraphs. For comparison, we transform the multi-layer network into a single-layer network, and use layer information as attribute input.

**(v) MultiSup** [41] extracts a set of elaborate structural representations of links from all layers.

**(vi) MADM** [27] treats the combination of information from different layers in a multi-layer network as a multi-attribute decision-making problem, utilizing resource allocation metrics to compute intra-layer similarity and cosine similarity to calculate inter-layer similarity.

**(vii) MANE** [6] treats each layer as a distinct "view" and leverages two core principles, "diversity" and "collaboration," to enhance representation learning.

**(viii) MNERLP** [32] calculates node and edge relevance using the local and global representations, and combines both factors to perform link prediction.

**(ix) HOPLP** [33] combines the multi-layer network into a single weighted network while accounting for the relative density of layers, and then iteratively calculates link likelihoods by considering longer paths between nodes.

These methods represent a spectrum of approaches in multi-layer network analysis, each leveraging different strategies to predict links across multiple layers. By comparing LUSTER against these baselines, we aim to demonstrate the efficacy and advancements of LUSTER for link prediction in multi-layer networks.

## A.4 Evaluation Metrics

In this section, we provide detailed explanation and mathematical formulation about the evaluation metrics.

**(i) Accuracy (Acc).** Accuracy measures the percentage of correctly predicted samples out of the total. It provides a straightforward indication of overall predictive correctness. It is calculated as:

$$\text{Accuracy} = \frac{TP + TN}{TP + TN + FP + FN} \quad (16)$$

where TP (True Positive) is the number of true positive predictions. TN (True Negative) is the number of true negative predictions. FP (False Positive) is the number of false positive predictions. FN (False Negative) is the number of false negative predictions.

**(ii) Area under the ROC Curve (AUC).** AUC refers to the area under the Receiver Operating Characteristic (ROC) curve. The ROC curve illustrates the trade-off between the True Positive Rate (TPR) and the False Positive Rate (FPR) across different classification thresholds. True Positive Rate (TPR) measures the proportion of actual positive samples that are correctly identified as positive. The formula is:

$$\text{TPR} = \frac{TP}{TP + FN} \quad (17)$$

False Positive Rate (FPR) measures the proportion of actual negative samples that are incorrectly identified as positive. The formula is:

$$\text{FPR} = \frac{FP}{FP + TN} \quad (18)$$

The AUC value ranges from 0 to 1, where 1 indicates a perfect classifier and 0.5 indicates a model with performance equivalent to random guessing. This metric is particularly useful for evaluating binary classification models. A higher AUC value indicates better discrimination ability between positive and negative samples.

In general, the higher the Accuracy and AUC values, the better the model performance. These metrics collectively provide a comprehensive assessment of how well the models perform in link prediction tasks across various real-world datasets.

