# OpenReview forum: "LUSTER: Link Prediction Utilizing Shared-Latent Space Representation in Multi-Layer Networks"
_ACM.org/TheWebConf/2025/Conference — WWW 2025 Poster_

### Official Review · Reviewer_myNQ · 2024-11-26

**Novelty:** 5
**Technical Quality:** 5

**Review:**

### Evaluation of the Paper: *LUSTER: Link Prediction Using Shared Latent Space Representation in Multi-Layer Networks*

#### **Quality**
1. **Writing Style**: The paper is written in a formal academic style, with well-organized arguments and strong logical flow. It employs rigorous theoretical analysis and experimental evidence to support its claims.
2. **Structure**: The structure adheres to standard academic paper formats, including sections such as Introduction, Related Work, Methodology, Experiments, Discussion, and Conclusion.
3. **Technical Depth**: The usage of formulas, algorithms, and visualizations is clear and helps convey the key concepts effectively.

#### **Clarity**
1. **Text Clarity**: The text is clear, and the language is precise, making it suitable for researchers in the field.
2. **Use of Figures and Tables**: Figures and tables provide intuitive support for understanding experimental results.
3. **Formulas**: Key evaluation metrics such as accuracy and AUC (Area Under Curve) are presented with clear definitions.

#### **Originality**
1. **Innovation**: The paper proposes a novel method, LUSTER, that leverages shared latent space for link prediction in multi-layer networks. This approach demonstrates originality in multi-modal data representation and learning.
2. **Comparison with Baselines**: The method is benchmarked against existing approaches, showing competitive advantages in metrics like AUC and accuracy.
3. **Theoretical Contribution**: The shared latent space concept extends possibilities for modeling complex multi-layer networks.

#### **Significance**
1. **Research Significance**: Link prediction in multi-layer networks is a critical task in network science and social network analysis, with applications in recommendation systems, transportation analysis, and more.
2. **Practical Value**: By effectively integrating multi-modal features, LUSTER has the potential to improve efficiency in industrial applications.
3. **Experimental Evidence**: The method’s robustness and generalizability are validated across multiple real-world datasets.

---

### **Strengths**
1. **Methodological Innovation**: LUSTER utilizes shared latent space to unify feature representation across different network layers, offering a simple yet powerful approach.
2. **Comprehensive Experiments**: Extensive experiments on multiple datasets with detailed analysis of parameter sensitivity and hyperparameters.
3. **High Academic Value**: The study aligns with recent trends in multi-modal learning and graph neural networks, contributing to the advancement of the field.
4. **Clear Formula Derivation**: Complex formulas are explained with clarity, making them accessible to readers.
5. **Cross-Domain Applicability**: The method is not limited to social networks and can be generalized to other multi-modal network scenarios.

---

### **Weaknesses**
1. **Complexity Analysis**: Although the method is claimed to be efficient, it lacks a detailed analysis of time and space complexity.
2. **Scalability Concerns**: The applicability of the method to very large-scale networks is not thoroughly discussed.
3. **Limited Baseline Comparisons**: While some popular baselines are included, the comparison could be broadened by incorporating more recent approaches.

**Questions:**

1. Can you provide a detailed analysis of the computational complexity of LUSTER? Specifically, how does it scale with the number of layers and nodes in the network? How does the runtime and memory usage compare to baseline methods on larger datasets?
2. How do you ensure that the shared latent space representation effectively captures both layer-specific and global features in multi-layer networks? Have you explored whether adding constraints or regularization terms could improve the disentanglement of features between different layers?
3. How does LUSTER handle noise or missing data in one or more layers? Have you evaluated its robustness in such scenarios?

**Reviewer Confidence:**

2: The reviewer is willing to defend the evaluation, but it is likely that the reviewer did not understand parts of the paper

**Scope:**

4: The work is relevant to the Web and to the track, and is of broad interest to the community

---

### Official Review · Reviewer_zsbY · 2024-11-30

**Novelty:** 5
**Technical Quality:** 4

**Review:**

This paper proposes LUSTER, a new link prediction algorithm for multi-layer networks based on shared-latent spaces. LUSTER integrates intra-layer representations and shared-latent space representations (the latter obtained via adversarial training) for enhancing LP performances, achieving a notable improvement in AUC w.r.t. competing methods.

*Strengths*
- The core idea of using a combination of the shared latent space and the layer-specific one (without redundancies) is interesting and novel.
- The manuscript is well written written and easy to follow.
- The obtained results are good, as LUSTER always overcomes competing methods across datasets.
- The ablation study confirms the robustness of the proposed approach.
- The paper seems to foster reproducibility, as all the needed information are reported clearly. Furthermore, the authors share code and data (in an anonymous repo).

*Weaknesses*
- The authors claim that existing LP methods do not fully incorporate the role of the coupling in multilayer networks. However, [1] does this, considering each pair of layers in the link prediction. Furthermore, despite the approach proposed in [1] achieves remarkable results, it does not figure within related work nor experimental setup as a competing one.
- Related to this, there are some other notable multilayer link prediction methods that are missing, as in the case of [2, 3].
- The motivating Fig. 1 is not convincing to me, as it does not add something to the "reasons" behind LUSTER, remaining very generic in motivating the idea of LP on ML networks.
- The representation extractor module in Sect. 4.1 is a "simple" GCN. Using other architectures such as GAT or a SoTA GNN method for LP (e.g., [4]) for obtaining more powerful representations has not been explored, yet could lead to performance improvements.
- The "h consecutive links" concept mentioned in lines 380-381 is not clear to me, the authors should be more precise here and better explain how this contributes to learning a space representative for interconnections among different layers.
- In Algorithm 1, pe for unobserved edges is computed outside the actual training loop (rows 15-16 of the algorithm). However, they figure in Eqs. 5-9 (and row 6 of the algorithm), and therefore should be computed during each epoch.

[1] Zangari, L., Mandaglio, D., & Tagarelli, A. (2024, May). Link Prediction on Multilayer Networks through Learning of Within-Layer and Across-Layer Node-Pair Structural Features and Node Embedding Similarity. In Proceedings of the ACM on Web Conference 2024 (pp. 924-935).

[2] Cen, Y., Zou, X., Zhang, J., Yang, H., Zhou, J., & Tang, J. (2019, July). Representation learning for attributed multiplex heterogeneous network. In Proceedings of the 25th ACM SIGKDD international conference on knowledge discovery & data mining (pp. 1358-1368).

[3] Coscia, M., Borgelt, C., & Szell, M. (2022). Fast Multiplex Graph Association Rules for Link Prediction. arXiv preprint arXiv:2211.12094.

[4] Yun, S., Kim, S., Lee, J., Kang, J., & Kim, H. J. (2021). Neo-gnns: Neighborhood overlap-aware graph neural networks for link prediction. Advances in Neural Information Processing Systems, 34, 13683-13694.

**Questions:**

See weak points.

**Reviewer Confidence:**

3: The reviewer is confident but not certain that the evaluation is correct

**Scope:**

4: The work is relevant to the Web and to the track, and is of broad interest to the community

---

### Official Review · Reviewer_HvXS · 2024-12-02

**Novelty:** 4
**Technical Quality:** 4

**Review:**

**Paper Summary:**

The paper addresses the problem of link prediction in multi-layer networks, emphasizing the need to model shared-latent spaces for better inter-layer information sharing. The proposed framework, LUSTER (Link prediction Utilizing Shared-laTent spacE Representation), combines intra-layer and shared-latent space representations using four modules: Representation Extractor, Latent Space Learner (with adversarial training), Complementary Enhancer (using orthogonal fusion), and Link Predictor. Extensive experiments demonstrate LUSTER's effectiveness, achieving significant improvements in AUC over state-of-the-art methods.

**Summary of Strengths:**

+ The introduction of a shared-latent space with adversarial training is a creative and innovative solution for capturing inter-layer coupling in multi-layer networks.

+ The experimental setup is robust, demonstrating consistent and significant performance improvements (up to 15.87% AUC gain) over state-of-the-art methods.

**Summary of Weaknesses:**

+ The representation extractor of LUSTER utilizes 𝐾 separate (GCN)  to learn and/or extract the intra-layer representations of each individual layer. The paper lacks sufficient discussion on the computational complexity  of the LUSTER framework.

**Questions:**

none

**Reviewer Confidence:**

1: The reviewer's evaluation is an educated guess

**Scope:**

3: The work is somewhat relevant to the Web and to the track, and is of narrow interest to a sub-community

---

### Official Review · Reviewer_xZdo · 2024-12-03

**Novelty:** 5
**Technical Quality:** 5

**Review:**

The paper presents LUSTER, a novel framework for link prediction in multi-layer networks. Unlike existing methods that merge networks or rely on similarity metrics, LUSTER focuses on leveraging the shared-latent space between layers to enhance inter-layer information sharing. Overall, the proposed solution is sound and reasonable.

Pros:
1. The paper presents a well-structured, innovative approach to link prediction in multi-layer networks. The authors effectively address the limitations of existing methods. The use of shared-latent space representation enhances the coupling between network layers, improving link prediction accuracy.
2. Extensive experiments on various real-world datasets demonstrate the effectiveness and robustness of LUSTER.
3. The paper is well-written and easy to follow. It uses illustrative figures and visualizations to effectively present the main ideas and results.

Cons:
1. The research novelty is not very strong. Link prediction has been extensively studied and there are numerous relevant studies. More explanation is needed to highlight the novelty of this work.
2. The motivation for utilizing the latent space is not clearly presented. The use of a generator-discriminator is not particularly new.
3. The paper lacks an analysis of time complexity or running time for the link prediction task.

**Questions:**

1. What are the unique benefits of this approach compared to other techniques, such as GNNs or other recent graph learning models?
2. In the experiments, why do the latest methods (e.g., MNERLP [32], HOPLP [33]) perform worse than the older methods?

**Reviewer Confidence:**

3: The reviewer is confident but not certain that the evaluation is correct

**Scope:**

3: The work is somewhat relevant to the Web and to the track, and is of narrow interest to a sub-community